# Gaming Preferences and Personality among School Students

**DOI:** 10.3390/children10030428

**Published:** 2023-02-22

**Authors:** Balan Rathakrishnan, Soon Singh Bikar Singh, Azizi Yahaya

**Affiliations:** Faculty of Psychology and Education, University Malaysia Sabah, Kota Kinabalu 88400, Malaysia

**Keywords:** personality, gaming, school student, gender

## Abstract

Gaming has vastly developed into numerous genres; nonetheless, most studies in the literature emphasize the violent genre only. Therefore, this study was conducted to investigate the relationship between personality and gaming preferences among school students. This study also aims to investigate the difference in the time spent on gaming based on gender. The third objective involves examining the differences in frequency in playing a video game based on age. The sample group comprised 420 school students aged between 12 and 17 years old, including hardcore and casual gamers. The online survey was conducted using Google Forms, and the participants were requested to answer the demographic questionnaire, Big Five Inventory, and Gaming Preferences Questionnaire. The obtained data were analyzed using SPSS 26.0 software for general descriptive statistics. The results show that there was a significant relationship between gaming preferences role-playing game (RPG), combat, online, and music genres) and personality (extraversion, agreeableness, conscientious, neuroticism, and openness). The results also indicate a difference between gender and the number of hours spent on gaming, but no such difference existed between age and the frequency of playing games. From the psychological perspective, gaming preference is related to their personality and influences the behavior of children and overall health in daily life.

## 1. Introduction

Out of about 20.1 million video game players in Malaysia, more than half of them are school students and young adults ranging from 13 to 25 years old [1]. Video games have long been associated with school students as they have always been the primary consumers of the industry ever since it was popularized in the early 1980s [2]. The decision of someone to involve any appropriate positive or negative behavior is related to their situational element such as their mood and their consistent elements such as personality [3]. This shows that someone who is depressed more enjoys video games with darker undertones than someone who feels happy. Someone interested in some games and activities could be their personality [3].

Based on ESA in 2019, 40% of video game players are comprised of adolescents and young adults aged from 12 to 18 years old. This age group has the highest hours reported in spending their time playing video games with an average of 11.89 hours each week (Limelight Networks, 2020). Older gamers (>35 years old) consider themselves to be more of a novice or casual gamers who, on average, would only spend about 8.64 hours per week, while aspiring professionals and experts were said to be more from the younger crowd where they would spend about 17.58 hours a week playing video games. This would be due to the fact of the games that these age groups prefer. The amount of time school students spends playing video games has previously been proven to be influenced by their preferred genre [4]. 

Gaming preferences have always been a subject of concern for parents who have children and school students going through their growth phases [5]. This is due to the stigma that gaming preferences, especially those that focus on violence, would affect how their child would presume the world is and shape their behavior to mimic the ones they see on their screens [6]. As gaming preferences keep improving in aspects of their visual detail and realism, the American Psychological Association in 2015 stated there is a correlation between the use of violent gaming preferences and aggressive behavior [7].

Nevertheless, gaming preferences video games playing do not only bring out the negative in people. It is very beneficial to a person’s social and cognitive development and psychological well-being [8]. Players who play the action genre of gaming preferences video games playing have been said to have better hand–eye coordination and visuomotor skills, such as resistance to distraction, sensitivity to information in the peripheral vision, and the ability to count briefly presented objects, than those who do not play games [7,9]. The development of gaming preferences video games playing through PlayStation Move and Nintendo Switch has helped players develop motor skills through whole-body movement [10]. It was also found out that gaming preferences video games playing increase the visual and attentional skills as players were able to track two more objects on average than non-players [11]. Still, they were more likely to recognize targets in a cluttered area [12]. With enough research on this issue, it could be that video game playing brings more benefits to people than what is generally known.

Personality had had its fair share on interpretation in correlation with video games. Yet, most of the research has only been focusing on aggression and violence in the games themselves. One of the earliest studies that show some understanding of video games and their effects on their players [13], where they had examined the differences in personality between groups of higher and lower video game usage. There were no personality distinctions found in between the two groups by the dimensions that were studied, which were self-esteem/degradation, social-deviance/conformity, hostility/kindness, social withdrawal/gregariousness, obsessive-compulsive, and achievement-motivation. 

Studied the differences in personality between people who play video games and those who do not. The study circled around seven personality and behavioral variables which were then identified as neuroticism, extraversion, psychoticism, honesty, antisocial behavior, criminal behavior, and gambling. Extraversion was the only trait that had a significant difference between the two groups where people who play video games were higher in that trait. It was then concluded that there was not a strong personality score to differentiate people who play video games and people who do not play them [6,14].

This would be good combat to trait personality theory, which explains that the personality of an individual is divided into five main traits [15,16,17]. The Big Five personality traits are extraversion, agreeableness, openness, conscientiousness, and neuroticism. Each trait represents a continuum. Individuals can fall anywhere on the continuum for each trait. The Big Five remain relatively stable throughout most of one’s lifetime. This theory says that people selectively expose themselves to media based on the drives and emotions they feel when they are in gameplay. Studies show that most school students who are addicted to computer games have a high heartbeat and blood pressure due to too much excitement and stress [18,19].

The other aspect is video games have always and will always be thought of as a male-dominated media. This has been heavily supported by past research such as those studies by [20] Both studies concluded that men enjoy playing video games more than women do. This would be due to the social view of video games that is looked upon as a masculine activity and if females were to continue playing it, then they will lose the feeling of inclusion and affection that they crave from their other female peers [20] However, with the 21st century and entering a new decade, the direction has taken a turn. Female video game players have significantly increased in the past few years. 

The Entertainment Software Association (ESA) reported that in 2019, 46% of the gamers consisted of females. Yet even with that, it was found out that men play video games for a significantly longer time in their lifetimes than women, where men have an average of ten years while women are at the average of two to five years [12]. Men play nearly seven hours per week which is an hour and two minutes longer than women who play about 5.80 hours [21]. It was found that women spend less time to play video games than men because they are required to fulfill more obligatory activities which leaves them to have less available leisure time and makes them less likely to “make” time for video gameplay [22] Both men and women play video games, but it was found that there is a difference in platform and game genre preference [23] ESA stated that males ranging from 12 to 18 years old enjoy playing genres compromised of action, shooters, and sports games, while women in the same age range enjoy genres such as casual and action games.

Male video game players are likewise considered to outnumber female video game players by a large margin. Males regularly like and play video games more than girls, according to research [9,10]. This trend, however, appears to be waning. Females make up a large percentage of video game players, according to current figures. According to a survey issued by the ESA in 2011, 40 percent of gamers are female [11]. In contrast, some studies have indicated that the percentage of female gamers is far smaller. Terlecki et al. (2010) showed that just 27% of female participants, compared to 74% of male participants, said they were presently playing video games at the time of the research [12]. Research also discovered that males reported playing video games for ten years on average, compared to two to five years for girls [12]. This indicates that, on average, girls spend less time playing video games than guys, yet they still participate. As a result, males spend more time playing video games on average than females.

Research on this topic has always been inadequate. Most of the past research has focused only on the priorities for violent gaming preferences, which is inadequate to meet the needs of the gaming community. It is necessary for this research to be conducted in examining the association between a person’s personality and their preferences in video gameplay. Therefore, the objectives of this study are (1) to investigate the association between personality and gaming preferences in school students; (2) to investigate the differences between gender and the hours spent on gaming in school students; and (3) to examine the differences between age and the regularity of gameplay among school students. 

## 2. Materials and Methods

### 2.1. Research Design

The study investigated the relationship between gamers’ personality and their gaming preferences. The research method was survey-based. All participants completed three self-administered questionnaires during a single session. The first questionnaire collected the respondent’s background, i.e., age and gender. 

While online surveys using list-based sampling frames can be conducted either via the web or by e-mail, if an all-electronic approach is preferred, the invitation to take the survey will almost always be made via e-mail. Additionally, because e-mail lists of general populations are generally not available, this survey approach is most applicable to large homogeneous groups for which a sampling frame with e-mail addresses can be assembled (e.g., universities, government organizations, large corporations, etc.). This is calls these ‘list-based samples of high-coverage populations.

The second questionnaire was on gaming preferences, i.e., a comprehensive list of game elements that would assess whether the players played the games based on the games’ characteristics. The third questionnaire was the personality test, Big Five Inventory (BFI). It examined the respondents’ personality profile based on the five-factor model, and the results would then produce their respective scoring in terms of extraversion, agreeableness, conscientiousness, neuroticism, and openness.

Before collecting the data, the research obtained approval from REB (4Ketika 4/20(11)). The first part of the questionnaire consists of confidentiality, and consent has been obtained before all subjects could participate in this study. The age group is underage, so the parents and guardians gave written approval to conduct this research. 

If the subject is not comfortable answering, they can withdraw anytime from the studies. Ethical issues have been taken care of before starting to give the questionnaires to respondents. 

### 2.2. Research Sample

The population of this research consisted of school students (at least 12 years old), including both hardcore and casual gamers. As it was nearly impossible to obtain a probability sample, some considerations were considered regarding the sampling of this study.

According to the Chief Statistician, Datuk Seri Dr Mohd Uzir Mahidin said that the number of children in Sabah is around 1.02 million, which is an age group lower than 18 years. The overall target number of respondents for this research was in the hope of collecting 500 students. Instead, it settled with having 420 students as its respondents. The number was considered reasonable, which was still being able to collect sufficient research data while amid limited network coverage issues for some students who live in rural areas of Malaysia. According to Krejcie and Morgan [24], the minimum sample size should be 317 [17], but university students were randomly selected until the number of respondents reached 420 students.

Subjects of this study should be game-literate players since they would be more likely to have a broader gaming experience and be able to choose their own preferences. The risk with including low or no gaming exposure participants was that they could only select a few of those gaming elements that they had tried or been told about, categorized as a selection of preferences. Another consideration was that the participants were self-selected, in which they chose to partake in the study when they saw the announcement made via WhatsApp. 

Therefore, the generalization of the results would be considered. The final consideration was the use of the Internet as a medium for collecting data. This research seeks a nationwide outreach, and the Internet stands out as a good platform. Accessibility to the Internet is not a sampling bias as gamers are the most technologically inclined people. The survey itself was hosted on Google Forms for easier access to the respondents.

### 2.3. Location of Study

The data were collected from tertiary education institutes in the states of Selangor, Sabah, and Sarawak in Malaysia. 

### 2.4. Sampling Technique

Convenient sampling was used as the entire sampling process was carried out in a single step, with each subject independently selected from the other population members. This sampling technique demonstrated good generalization power of the findings to the entire population. A total of 420 school students were recruited in this study. It showed a satisfactory sample size as the population of Malaysian school students is recorded at approximately 5,000,000 (age between 12 and 17) [24]. In other words, this sampling method involves obtaining participants wherever they can be found and typically wherever is convenient. In convenience sampling, no inclusion criteria are identified prior to the selection of subjects. All subjects are invited to participate.

In this study, the questionnaire has been distributed to the schoolteachers who are willing to distribute the questionnaire willingly. The researcher distributed the questionnaire through existing contacts. The researcher sends the online questionnaire to individuals on their mobile phone’s contact list, connected to via social networking websites such as Facebook, LinkedIn, and Google+ and to individuals whom the researcher knows in person. This would be the easiest and the most convenient way of recruiting the sources of the primary data for this research.

### 2.5. Data Gathering Procedures

The data were collected from June to August 2020. An announcement calling for participants was sent out to diverse groups via WhatsApp and iMessage. The participants completed the online survey from their computers or handphones at their own convenient time. The survey tool was designed not to allow the participants to save their progress; therefore, the participants had to complete the whole survey in one session.

The online survey was set up using Google Forms, whereby the participants were presented with the demographic questionnaire. Then, they were given the Big Five Inventory and Gaming Preferences Questionnaire and were thanked for their participation.

### 2.6. Research Instrument

#### 2.6.1. Respondent Background

This section consists of questions such as gender, age, ethnicity, time spent on video games once started, and the frequency of playing the video game. 

#### 2.6.2. Big Five Inventory (BFI)

Big Five Inventory by John and Srivastava [25] is a 44-item measure that yields a score for each of the Big Five personality factors: extraversion (eight items), agreeableness (nine items), conscientiousness (nine items), neuroticism (eight items), and openness to experience (10 items). Each item consisted of a short statement, and respondents were required to rate the degree to which they would agree to each statement on a 5-point Likert scale (1 = “Strongly Disagree” to 5 = “Strongly Agree”). Finally, a fairly acceptable Cronbach’s alpha (α = 0.67) was recorded for the BFI. Similarly, the dimensions of BFI recorded a range between 0.56 and 0.87 of reliability. The five personalities been classified as conscientiousness (impulsive, disorganized vs. disciplined, careful) agreeableness (suspicious, uncooperative vs. trusting, helpful), neuroticism (calm, confident vs. anxious, pessimistic), openness to experience (prefers routine, practical vs. imaginative, spontaneous), and extraversion (reserved, thoughtful vs. sociable, fun-loving). For this research, the researchers have used the mean score for each personality trait.

#### 2.6.3. Gaming Preferences Questionnaire (GPQ)

Zammitto constructed the Gaming Preferences Questionnaire to empirically measure video game players’ preferences for different popular video game genres [26]. In creating the questionnaire, she consulted six professional game designers who were asked to review the items and provide suggestions to improve the scale. All of the experts agreed on approving the questionnaire as an appropriate tool for measuring video game preferences. Zammitto’s study had the participants indicate their top three favorite games and favorite game genre [26]. The study randomly sampled 10% of participants’ data and compared their top three favorite games and their indicated favorite genre with the top three genres to indicate their Gaming Preferences Questionnaire scores. The questionnaire could predict the participants’ favorite genres in 91% of the 55 cases sampled. The questionnaire was then broken down into 8 main components that would make up a game genre: role-playing games (17 items), combat (11 items), puzzles (6 items), online (4 items), music (5 items), simulation (6 items), racing (7 items), and sport (5 items). A five-point Likert scale ranged from 1 = “strongly disagree” to 5 = “strongly agree”. The reliability test reveals the GPQ recorded excellent reliability (α = 0.95). 

The NEO-Five Factor Inventory (NEO-FFI) [5] was used to correlate reported video game type preferences with respondents’ five factor model (FFM) personality characteristics.

### 2.7. Data Analysis

The obtained data were analyzed using SPSS 26.0 software for general descriptive statistics, including age range, gender, ethnicity, how often they typically played games, how many hours they spent playing games, and how old they were when they first played games. Inferential statistics were used via independent sample t-test. The significant gender difference in how many hours they spent once they started playing games to answer the first research question was analyzed by applying the *t*-test. The significant differences between age and how often they typically played games and the meaningful differences between personality and gaming preferences, which were to answer the second and third research questions, were analyzed using the significance level of (*p* ≤ 0.05), which was opted in this research.

## 3. Results

### 3.1. Background of Respondents

Table 1 presents the background information and personality distribution of the 420 respondents in this study.

Table 2 shows the hours spent, frequency, and the age they started video gaming of the 420 respondents in this study.

### 3.2. Inferential Analysis

#### 3.2.1. Personality and Gaming Preferences

Table 3 presents a correlation analysis between personality and gaming preferences. The genres of RPG recorded a positive association (r = 0.235; *p* < 0.05) with the personality of conscientiousness. The genres of combat showed a positive association (r = 0.188; *p* < 0.05) with the personality of openness. The results revealed that the personality of extraversion positively correlated (r = 0.250; *p* < 0.05) with the genres of online. Moreover, the genres of music reported a positive relationship (r = 0.207; *p* < 0.05) with the personality of agreeableness.

#### 3.2.2. Gender Differences in the Hours Spent for Gaming

Ho2: There is no significant difference between gender and the hours spent on gaming in school students.

Table 4 presents the gender differences in the hours spent for gaming. An independent sample t-test was conducted to determine the gender differences in the hours spent for gaming among the sample of school students (*n* = 420). On average, males (M = 2.63, SD = 0.90) and females (M = 2.22, SD = 0.82) differed significantly in the hours spent for gaming; t (418) = 2.83, *p* = 0.005. Therefore, the null hypothesis, whereby there is no significant difference between gender and the hours spent for gaming preferences in school student, was rejected.

#### 3.2.3. Age and Frequency of Gameplay

Ho3: There is no significant difference between age and frequency of gameplay among school students.

Table 5 shows a one-way ANOVA conducted between the respondents’ age and the frequency of their gameplay. Since *p* < 0.05, it could be concluded that there was no significant difference between age and the frequency of the respondents’ gameplay F (3,416) = 0.98, *p* = 0.41. Therefore, the null hypothesis, which stated that there is no significant difference between age and the regularity of gameplay among school students, is not rejected.

## 4. Discussion

### 4.1. Personality and Gaming Preferences

#### 4.1.1. Conscientiousness and RPG

Based on the RPG genre, there was a statistically significant relationship between the personality of conscientiousness and RPG games (r = 0.235; *p* < 0.05). This was seen in Zammitto’s study, whereby she found that the players scored higher in conscientiousness in this genre because they preferred games with imaginative scenarios and rich narrative content for them to explore, which is RPG games [26]. It is intellectually challenging throughout the whole gameplay.

People who are more inclined to conscientiousness experiences feel satisfied in playing RPG games because of the number of challenges and issues that need to be resolved. Conscientious persons are hardworking, disciplined, pedantic, and they devote a lot of time to organization. These are persons who are intrinsically motivated and who make a lot of effort to be successful in what they do. This is related to their role play game, whereby conscientious persons are oriented towards goals and the execution of a task.

Peever, Johnson, and Gardner also discovered a positive correlation between conscientiousness and RPG [27]. The players were more likely to reflect their curious minds and exploratory nature as they played games like this. Those who enjoyed playing RPG games also scored higher in conscientiousness. Jeng and Teng’s study stated a high correlation between the RPG gaming genre and conscientiousness related to escapism motivation [28]. Escapism means the mental diversion from unpleasant or dull aspects of one’s life through activities that usually involve imagination or entertainment [29,30]. They are more geared toward playing RPG games, as it gives them a sense of escape from the real world by building and experiencing things that they cannot usually do, which provides them with a sense of accomplishment.

#### 4.1.2. Openness and Combat

There was a statistically significant relationship between the openness and combat (r = 0.188; *p* < 0.05). Chen, Tu, and Wang found similar results, whereby those who scored higher in openness enjoyed combat games because they were curious [31]. Therefore, these types of games that have unique effects or intellectual development become more appealing. They also stated that openness people prefer combat types of games because they are intrinsically motivated and make a lot of effort to succeed in their goal [28]. Zammitto also found the combat genre to be positively correlated with openness, which would mean that people in this category have good organizational skills and clear strategies and can identify sub-objectives [26].

Furthermore, Zammitto discovered that combat games were closely related to personalities such as openness [26]. As combat games are usually the ones that enter the players into a situation where tension, impulsion, and low patience are engaged in the game, openness would be the best definer for the preference in the genre. People who tend to be high in the trait of openness are more willing to embrace new things, fresh ideas, and novel experiences. They are open-minded and approach new things with curiosity and tend to seek out novelty, while individuals who are very low on the trait of openness are often seen as being rigid and close-minded.

It would explain why openness is also a trait that is supposed to be found in those who enjoy this genre. They are more likely to be aggressive, addicted to excitement, and decisive when they play according to gaming preferences, which is a potential explanation of different personality behaviors and lifestyle choices. Persons with a high score in this dimension are creative, imaginative, and since they have a broad range of interests, like to explore the unknown, while persons with low scores in this dimension are of conventional appearance and behaviors, narrowed interests, prone to conservative attitudes, and tend to prefer what is already known in relation to the unknown. This could be related to why those who are openness are more interested in combat games.

#### 4.1.3. Extraversion and Online

The online game genre and personality results were extraversion statistically significant related to online games (r = 0.250; *p* < 0.05). Many online games require quick decisions and good strategies for gameplay as the player is playing with many other players at the same time. Persons with a high score on extraversion are more open, more persistent, more talkative, and more social than those with a low score on extraversion, who are shy, quiet, and aloof. Extraversion is associated with values of achievement and hedonism, but also with goals relating to an exciting lifestyle. Peever, Johnson, and Gardner and Teng stated that extraversion is a big trait in those who play online games as these individuals are the ones that are creative, broad-minded, and intelligent, which help in mastering new skills while enhancing their capabilities [27,29].

Zammitto [26], on the other hand, found that there was a negative relationship between openness and the online genre, which was also tied to agreeableness. These types of games involve a high level of competition that would correlate to low agreeableness in that sense and mastering the game means that the player would need to know the situations that lower openness [26]. She also discovered that extraversion goes hand in hand with online games. These games connect people from all over the Internet, which is enabled by textboxes or voice chats. Extroverted people tend to be more talkative as they enjoy being in social activities that allow them to be surrounded by other people. Therefore, going and playing games online would draw in extroverts due to those traits.

#### 4.1.4. Agreeableness and Music

Correlation analysis determined a statistically significant association between agreeableness and music games (r = 0.207; *p* < 0.05). It was seen that those who enjoyed the music genre scored higher in agreeableness. Pasinski, Hannon, and Snyder also found similar results in their study, whereby gamers scored higher in agreeableness [32,33]. People who score high on agreeableness tend to be imaginative, have a wide range of interests, and are open to new changes in their environment.

These people have an aesthetic appreciation for the music being played in games, which promotes their gameplay. Agreeable people are usually kind, cooperative, and sympathetic. For this reason, they would relate themselves to the music they hear in the games they play more than other people. Agreeableness is a predictor of the emotional intensity experienced from all types of music, both positively and negatively [32]. Those who prefer the music genre in games could display an intense emotional response to music that they had never heard before and continue to enjoy it henceforth.

Peever, Johnson, and Gardner discovered another trait, extraversion, closely related to those who enjoyed the music genre [27,32]. Highly extroverted people are social, outgoing, and energetic, and they require higher levels of social interaction and movement, which is why they are more attracted to music games [34,35,36].

### 4.2. Gender Differences in the Hours Spent for Gaming

Regarding the independent sample t-test results carried out, it suggested that there was a significant difference between both female and male respondents’ scores in the hours taken for them to play games. The past findings from Dorgan [37] and Ivory [38] and had the same results, whereby men did significantly play more games than women.

Jansz, Avis, and Vosmeer [39] stated that people play games with different intentions and motivations that can be broken down into a few categories, such as achievement, power gain, competition, character optimization through game mechanics, the social aspects of gaming such as relationships, teamwork, and socializing in general, and immersion that comes from character customization, role-playing, escapism, and discovery. Fulfilling these needs of motivation is the drive that people feel when they play games. As a whole, Olson [40,41] specified that school students are motivated to play games due to the thrill-seeking behavior that the game can offer and the feeling of relaxation and positive mood on the other hand. Olson, Kutner, and Warner [41,42] found that boys preferred playing games that catered to violence as it gave them the means to express fantasies of power and glory and that they could explore and master what they perceived as an exciting and realistic environment. It was also seen that boys exhibit a greater need to relate awards with their success in gaming, which would be why they spend a lot more time playing games than girls do [38]. A study by Ruzic-baf, Strnak, and Debeljuh [43] also found that boys play more games than girls because boys search for social interaction in virtual life more than girls do. On the other hand, girls also use games to cope with their anger and other emotions. Hartmann and Klimmt discovered that women go for richer games in social interactions and dislike the heavy gender stereotyping in character presentation and violent content [3]. Bonanno and Kommers [44] and Rathakrishnan et al. [45] support this, whereby they found that women prefer games geared to the puzzle, adventure, fighting, and managerial genre, which makes up for their need for challenge and arousal.

### 4.3. Age and Frequency of Gaming Preferences

The results from the correlation test showed that age and the frequency of gameplay among school students were not significant F (3,146) = 0.98, *p* = 0.41. It suggested that whether the respondents were older or younger, it did not affect their gameplay whatsoever.

There is no specific research that breaks down the difference in age and frequency of gameplay specifically. Nevertheless, the Entertainment Software Association [46] found that it was true, whereby most gamers were in the category of 13 years old. Still, the numbers did not decline as drastically as one would expect. Limelight Networks [21] found that people aged 13–15 years old played at least 7.78 hours per week, while those aged 26–35 played 8.21 hours per week and those aged more than 35 years old only played less than 7 hours per week. It was seen that there was no significant difference in their playing time by different ages. Brown discovered that younger gamers had more time playing games because of the decline in work hours among them. In contrast, those who worked full- or part-time are about as likely as those who are unemployed and looking for work in their time of playing games [47].

GameSparks [48] stated that younger gamers preferred competition, while older gamers were not as into that aspect of gaming as much. On the other hand, strategy in gaming is a more stable motivation across ages, which is why games such as StarCraft II are famous among people across all ages [49]. Whitbourne, Ellenberg, and Akimoto [23,50] found that gamers aged 18–29 played games to “try to beat my friend and teammates” (47.7%), while those who were aged 30–39 years old played games for the “challenge” (24.2%) and “stress relief” (29.7%). The younger age group felt sharper and improved their memory by playing games than the older gamers, in which playing games made them better at seeing patterns and performing tasks more quickly [50,51].

Shi et al. [52] stated that gamers aged 16–35 years old played more games because gaming is meaningful and purposeful. It was described that games brought them friendships, became something they enjoyed, and wanted to work for the gaming industry themselves in the future on top of them constantly thinking about games. Another reason was a push and pull influence on the amount of time they spent on gaming. Personal, interpersonal, and environmental impacts pull them into games or push them away from it. The forces could be from an individual (intrinsic) or things that are irrelevant in their environment [50,51,52]. However, older gamers are not as motivated as they have more responsibilities and hobbies, which overpower their need to play gaming preferences. Many more casual gamers come from the older crowd as the level of design in games would be somewhat difficult for them to enjoy the game entirely.

As a limitation, the timeframe of the survey was not accurately documented, which is a restriction. It is impossible to say whether the pressures on students throughout their studies or the overall environment are the more stressful factors. Because the general conditions have already been passed for several weeks and, particularly towards the conclusion of the semester, there are additional tests, surveys at the end of the semester might generate more expectations than at the beginning. Responses in the sense of social wishful thinking are feasible in the case of paper surveys. Despite the presence of an interpreter, if there are language hurdles for overseas students, responses may be questionable. This study has the characteristics of a pilot study due to the limited sample size of school students. There should be more data collected [53,54]

Other than that, this research also does not focus on analysis between gender and video game playing, which could be good research in future.

The low sample size also could be one of the limitations. Similarly, it is difficult to infer a cause-and-effect relationship between type of play and personality, since, as mentioned, there are confounding factors that probably influence these relationships and have not been considered in this study (characteristics of the schoolchild). Therefore, these effects deserve to be further investigated in future studies and shed light on this association.

After receiving a fair idea about this bias using probability sampling, the researcher can use both convenience sampling and probability sampling techniques to draw a more accurate estimation. The probability of selection of samples along with the convenience sampling will increase the power to overcome the sample’s biases [55,56].

For upcoming research, research also can focus on the differences of personality and games they like to play.

## 5. Conclusions

This study will be of great help to an abundance of people ranging from children and adolescents, where it could give them the power to use video games to enhance their self-qualities and to refine the ones that are not beneficial to them, to parents and educators, where it would give them the opportunity to understand video games and see it in a different light so it can be used to their advantage in educating their children and the younger ones. The findings of this study can also grant the idea where video games are probable to be an asset and could reduce the social stigma of video games to the society, as video games itself is something that is growing to be a major part of entertainment of our ever-changing society.

The educational system should take responsibility for the development of students, including a brief review of educational legislation and the role of the institute to prevent and/or treat the type of problems derived from the type of game and the personality of the students. These are some suggestions that can be considered to improve the quality of education in this multicultural context and, especially of interest to teachers, guidance tips and, in general, members of the educational community since this is a journal of students, and considering that the article includes educational/social variables, I believe that a small effort can be made to broaden the educational implications of the study in the sense of what can be done in the future by education to improve the integral development of the students mentioned above.

After 3 months of data sampling and with a total of 420 respondents to study the relationship between personality and gaming preferences, this research revealed 3 significant outcomes: (1) There is a substantial relationship between gaming preferences [role-playing game (RPG), combat, online, and music genres] and personality (extraversion, agreeableness, conscientious, neuroticism, and openness). (2) There is a significant difference between gender and the hours taken to play games. (3) There is no significant difference between age and frequency of gameplay.

## Figures and Tables

**Table 1 children-10-00428-t001:** Descriptive analysis of respondents’ background (*n* = 420).

Respondents’ Background	*n* (%)
Gender	
Male	160 (38.1)
Female	260 (61.9)
Age	
12–13	132 (31.3)
14–15	168 (25.3)
16–17	120 (28.7)
Mean ± S.D.	1.97 ± 0.77
Ethnicity	
Malay	375 (89.3)
Chinese	11 (2.7)
Indian	3 (0.7)
Bumiputera Sabah	20 (4.7)
Bumiputera Sarawak	8 (2.0)
Others	3 (0.7)
Personality	e.g.,
Extraversion	3.22 ± 0.22
Agreeableness	3.17 ± 0.62
Conscientiousness	2.25 ± 0.21
Neuroticism	3.11 ± 0.36
Openness	3.40 ± 0.25

All demographic details related to gender, age, and ethnicity had been asked in part 1 of the questionnaire.

**Table 2 children-10-00428-t002:** Descriptive analysis of gaming patterns (*n* = 420).

Gaming Patterns	*n* (%)
Hours spent once start the game per day	
Less than 30 min	59 (14.1)
1–2 h	193 (45.9)
3–4 h	118 (28.1)
More than 5 h	50 (11.9)
Mean ± S.D.	2.64 ± 3.66
Frequency (Hours spent)	
Everyday	109 (26.0)
Almost everyday	115 (27.3)
Several times a week	79 (18.7)
Several times a month	48 (11.3)
Several times a year	25 (6.0)
Rarely	44 (10.7)
Age started video gaming	
5 years old or younger	64 (15.3)
Between 6 and 11 years old	255 (60.7)
Between 12 and 13 years old	70 (16.7)
Between 14 and 15 years old	31 (7.3)

**Table 3 children-10-00428-t003:** Correlation between personality and gaming preference.

	Extraversion	Agreeableness	Conscientiousness	Neuroticism	Openness
RPG	0.144	0.458	0.235 **	0.142	0.298
Combat	0.170	0.121	0.302	0.146	0.188 **
Puzzle	−0.031	0.146	0.250	0.121	0.458
Online	0.250 **	0.142	−0.056	0.458	0.302
Music	0.207	0.207 **	0.013	0.227	0.207
Stimulation	0.302	0.298	0.234	0.298	0.031
Racing	0.263	0.188	0.188	0.188	0.144
Sport	0.418	0.234	0.298	0.234	0.170

** *p* < 0.05.

**Table 4 children-10-00428-t004:** Gender difference of the hours spent for gaming.

Gender								
	Male		Female					
	M	SD	M	SD	df	t	95% CI for Mean Difference	*p*
Time taken to play games	2.63	0.90	2.22	0.82	418	2.83	0.12, 0.69	0.005 *

* *p* < 0.05.

**Table 5 children-10-00428-t005:** One-way ANOVA results between age and frequency of gameplay among school students.

	Sum of Squares	df	Mean Square	F	Sig.
Between Groups	7.60	3	2.53	0.98	0.41
Within Groups	377.76	416	2.59		

## Data Availability

Not applicable.

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
