# Peer review of "Gaming Preferences and Personality among School Students"

_children, 2023, doi:10.3390/children10030428_

Round 1

Reviewer 1 Report

General comment

The main lesson of the study presented in this article is to point out the existence of a link between some game preferences and some personality traits in a population of school students.

However, the purpose of this article is not entirely clear. The title suggests that it is a question of studying the links between gaming preferences and personality. In the abstract and the end of the introduction the objectives are stated:

1) to investigate the association between personality and gaming preferences in school student; 2) to investigate the differences between gender and the hours spent on gaming in school student; 3) to examine the differences between age and the regularity of gameplay among school students.

These objectives, too different, too numerous, which are not argued, make the whole lose a lot of coherence. It would have been better for the authors to focus on the first, possibly strengthening it by adding an analysis by gender, by crossing game preferences and gender. This arrangement would refocus the article on the most interesting subject.

The introduction deserves to be deeply revised with regard to a refocused objective. As it stands, it is difficult to pinpoint a common thread leading to the statement of purpose: How the topic of the article has been covered so far and why this topic is important. It also shows certain inaccuracies or developments that are difficult to understand (see specific comments).

In the methodological section, the description of the sample recruitment procedures should be enriched. The limitations in terms of representativeness of this sample should be discussed.

I suggest for the results to delete the two parts on “Gender Differences in the Hours Spent for Gaming” and “Age and Frequency of Gameplay”, and to add results of an analysis crossing gender and gaming preferences.

The discussion should include a section on the limitations of the study.

The conclusion should not go beyond the main findings of the study.

Specific comments:

• Line 28: “The decision of someone to involve any behavior is related to their situational element such as their mood and their consistent elements such as personality”

This sentence is too general (any behavior) when we are talking about the field of gaming and game preferences

• Line 36: “The NEO-Five Factor Inventory (NEO-FFI) [5] was used to correlate reported video game type preferences with respondents five factor model (FFM) personality characteristics”

This sentence has more place in the methods section

• Line 56: “As a result, while males spend more time playing video games on average than females, the sexes are more equivalent when it comes to who plays video games and who does not”

I do not understand what the authors mean. What are they relying on to say that?

• Line 59: “Nevertheless, gaming preferences do not only bring out the negative in people…”

Throughout this paragraph, it seems that the use of the term "gaming preferences" is not appropriate. Don't the authors rather want to refer more generally to gaming or video games playing?

• Table 1: Mean Age = 21.62 ± 3.94

How can we have such an average on a sample of 12-17 year olds?

• Table 2: Descriptive analysis of gaming preferences

The term gaming patterns rather than gaming preferences would be more appropriate because it is about game duration and frequency.

• Table 3

A caption is missing at the bottom of the table to explain the use of bold and the presence of **

Author Response

General comment

The main lesson of the study presented in this article is to point out the existence of a link between some game preferences and some personality traits in a population of school students.

However, the purpose of this article is not entirely clear. The title suggests that it is a question of studying the links between gaming preferences and personality. In the abstract and the end of the introduction the objectives are stated:

1) to investigate the association between personality and gaming preferences in school student; 2) to investigate the differences between gender and the hours spent on gaming in school student; 3) to examine the differences between age and the regularity of gameplay among school students.

These objectives, too different, too numerous, which are not argued, make the whole lose a lot of coherence. It would have been better for the authors to focus on the first, possibly strengthening it by adding an analysis by gender, by crossing game preferences and gender. This arrangement would refocus the article on the most interesting subject.

The introduction deserves to be deeply revised with regard to a refocused objective. As it stands, it is difficult to pinpoint a common thread leading to the statement of purpose: How the topic of the article has been covered so far and why this topic is important. It also shows certain inaccuracies or developments that are difficult to understand (see specific comments).

The main concern is about personality (iv) and gaming preferences(dv). And gaming preferences been checked by age and gender.

Personality had had its fair share on interpretation in correlation with video games yet most of the research has only been focusing on aggression and violence in the games itself. One of the earliest studies that show some understanding of video games and their effects towards their players was from Gibb et al. in 1983 where they had examined the differences in personality between groups of higher and lower video game usage. There were no personality distinctions found in between the two groups by the dimensions that were studies which where self-esteem/degradation, social-deviance/conformity, hostility/kindness, socialwithdrawal/gregariousness, obsessive-compulsive and achievement-motivation.

Estallo (1994) studied the differences in personality between people who play video games and those who don’t. The study circled around seven personality and behavioral variables which were then identified as neuroticism, extraversion, psychoticism, honesty, antisocial behavior, criminal behavior and gambling. Extraversion was the only trait that had a significant difference between the two groups where people who play video games were higher in that trait. It was then concluded that there wasn’t a strong personality score

Video games have always and will always be thought to be a male dominated media. This has been heavily supported by past research such as by Williams et al. in 2009 and Robinson et al. in 2008. Both studies concluded that men enjoy playing video games more than women do. This would be due to the social view of video games that is looked upon as a masculine activity and if females were to continue playing it, then they will lose the feeling of inclusion and affection that they crave from their other female peers (Lucas & Sherry, 2004). But with the 21st century and entering a new decade, the direction has taken a turn. Female video game players have significantly increased in the past few years.

Based on ESA in 2019, 40% of video game players are comprised of adolescents and young adults aged from 12 to 18 years old. This age group has the highest hours reported in spending their time playing video games with an average of 11.89 hours each week (Limelight Networks, 2020). Older gamers (> 35 years old) consider themselves to be more of a novice or casual gamers who, on average would only spend about 8.64 hours per week while aspiring professionals and experts were said to be more from the younger crowd where they would spend about 17.58 hours a week on playing video games. This would be due to the fact of the games that these age groups prefer.

In the methodological section, the description of the sample recruitment procedures should be enriched. The limitations in terms of representativeness of this sample should be discussed.

According to the Chief Statistician, Datuk Seri Dr Mohd Uzir Mahidin said that the number of children in Sabah around 1.02 million which age group lower then than 18 years. The overall target number of respondents for this research was in the hope of collecting 500 students, instead it settled with having 420 students as its respondents. The number was considered reasonable, which was still being able to collect sufficient research data while amid limited network coverage issues for some students who live in rural areas of Malaysia. According to Krejcie & Morgan, the minimum sample size should be 317 [18], but university students were randomly selected until the number of respondents reached 420 students.

I suggest for the results to delete the two parts on “Gender Differences in the Hours Spent for Gaming” and “Age and Frequency of Gameplay”, and to add results of an analysis crossing gender and gaming preferences.

The analysis crossing gender and gaming preferences was not in my objective, Maybe can be part of limitation

The discussion should include a section on the limitations of the study.

Other than that, this research also not focusing not focusing in analysis between gender and video game playing which could be good research in future.

The low sample size also could be one of the limitations. Similarly, it is difficult to infer a cause-and-effect relationship between type of play and personality, since, as mentioned there are confounding factors that probably influence these relationships and have not been considered in this study (characteristics of the schoolchild). Therefore, these effects deserve to be further investigated in future studies and shed lighter on this association.

The conclusion should not go beyond the main findings of the study.

Added

Educational system should take responsibility for the development of students, including a brief review of educational legislation and the role of the institute to prevent and/or treat this type of problems derived from the type of game and the personality of the students. These are some suggestions that can be considered to improve the quality of education in this multicultural context and, especially of interest to teachers, guidance tips and, in general, members of the educational community since this is a journal of students, and considering that the article includes educational/social variables, I believe that a small effort can be made to broaden the educational implications of the study in the sense of what can be done in the future by education to improve the integral development of the students mentioned above.

Specific comments:

  • Line 28: “The decision of someone to involve any behavior is related to their situational element such as their mood and their consistent elements such as personality”

This sentence is too general (any behavior) when we are talking about the field of gaming and game preferences.

The decision of someone to involve any appropriate positive or negative behavior is related to their situational element such as their mood and their consistent elements such as personality [3].

  • Line 36: “The NEO-Five Factor Inventory (NEO-FFI) [5] was used to correlate reported video game type preferences with respondents five factor model (FFM) personality characteristics”

This sentence has more place in the methods section

The NEO-Five Factor Inventory (NEO-FFI) [5] was used to correlate reported video game type preferences with respondents' five factor model (FFM) personality characteristics.

  • Line 56: “As a result, while males spend more time playing video games on average than females, the sexes are more equivalent when it comes to who plays video games and who does not”

I do not understand what the authors mean. What are they relying on to say that?

Some of the sentences been removed.

As a result, while males spend more time playing video games on average than females.  the sexes are more equivalent when it comes to who plays video games and who does not.

  • Line 59: “Nevertheless, gaming preferences do not only bring out the negative in people…”

Throughout this paragraph, it seems that the use of the term "gaming preferences" is not appropriate. Don't the authors rather want to refer more generally to gaming or video games playing?

Removed gaming preferences to video games playing

  • Table 1: Mean Age = 21.62 ± 3.94

Sorry for the typo error

Corrected to 1.97 (mean) and 0. 77 (S.D)

How can we have such an average on a sample of 12-17 year olds?

This statement been corrected

  • Table 2: Descriptive analysis of gaming preferences

The term gaming patterns rather than gaming preferences would be more appropriate because it is about game duration and frequency.

This statement been changed to gaming patterns

  • Table 3

A caption is missing at the bottom of the table to explain the use of bold and the presence of **

Reviewer 2 Report

The paper presents a research of interest in education at the secondary school stage. It has analysed the relationship between personality and gaming preferences of students, the difference in the time dedicated to gaming according to sex and the frequency of playing a video game according to age. It is a rigorous study, documented, explained and with relevant and recent references within the area.

The length of the paper seems adequate and conforms to what is proposed in the journal's instructions. The structure is also coherent and pertinent, clearly including theoretical aspects and previous studies in the introduction, methodological details of the research, results, discussion and conclusions. 

The title is appropriate and succinctly presents the topic addressed, as well as the abstract. In the case of the latter, a longer sentence is missing in the Methods section with a small description of the persons participating in the study (occasional players), which can be done in a single sentence.

The keyword "academic performance" does not refer to the study. I suggest that it be changed to a more accurate one. The rest of the words are well selected and represent the key issues addressed in the abstract.

Regarding the originality of the study, it is a topic covered in previous studies (which are not overly addressed in the theoretical framework of the text). However, as the authors indicate, it seems that this type of work focuses on other variables (other types of games), and the fact of using an appropriate methodology together with a secondary school sample in a current environment is a strong point of the study. Furthermore, it also seems to be a strong point and an original aspect of the work to analyse the relationship between the two aspects studied (types of games and personality) at this educational stage. This type of study with this sample is difficult to carry out, so it seems clearly a valuable aspect of this research.

As mentioned above, the state of the art on the subject is well constructed, although it includes few studies of relevant and recent quality in the field of research, both nationally and internationally. The design of the study is also rigorous, applying data collection instruments appropriate for the variables proposed to be studied and the possible relationship between the variables considered. The statistical analysis applied is correct and well justified. Regarding the results, they are well described and the categories help to organize the information on the measures used and the data obtained. However, it is suggested to add a more complete statistical analysis showing some predictive model between game type and personality. That is, how much the personality of the students is explained by the type of game they play.

Also, although some limitations are shown in the study, in my humble opinion other limitations are not adequately included. I suggest supplementing: "However, these findings should be interpreted with caution due to the fact that this study was not interventional, but was based on data reported by schoolchildren. Also, undoubtedly, the low sample size is another limitation. Similarly, it is difficult to infer a cause and effect relationship between type of play and personality, since, as mentioned there are confounding factors that probably influence these relationships and have not been considered in this study (characteristics of the schoolchild). Therefore, these effects deserve to be further investigated in future studies and shed more light on this association."

In addition, although the conclusions do not highlight the method of the selected study, they do briefly put forward some ideas on prospective that could be completed with some reflection and general documentation on educational strategies in this area. That is, it is suggested to briefly reflect on the extent to which the educational system should take responsibility for the development of students, including a brief review of educational legislation and the role of the institute to prevent and/or treat this type of problems derived from the type of game and the personality of the students. These are some suggestions that can be considered to improve the quality of education in this multicultural context and, especially of interest to teachers, guidance tips and, in general, members of the educational community since this is a journal of students, and considering that the article includes educational/social variables, I believe that a small effort can be made to broaden the educational implications of the study in the sense of what can be done in the future by education to improve the integral development of the students mentioned above.

Finally, it should be pointed out that the bibliographic references are adequate since they follow the standards required by the journal, both in the text itself and in the final references section. It is also suggested to review some spelling mistakes in the text (example, see Table 1).

PRESENTATION AND ADEQUACY TO THE JOURNAL'S STANDARDS

- Title: It is adequate since it is in accordance with the recommended norms.

- Abstract: It is of a length (221 words) adjusted to what is recommended in the norms.

- Key words: It does not adjust a word to the object of study.

- Length: Complies with the regulations.

- Structure: Conforms to standards.

- Names, symbols and nomenclature: Complies with standards.

- Schemes, drawings, graphs, charts, tables, etc.: Conforms to standards.

- Bibliographical references: Complies with standards. It is suggested to include a study from the children's journal related to the object of this study to complement the quality of the same. 

Author Response

The paper presents research of interest in education at the secondary school stage. It has analyzed the relationship between personality and gaming preferences of students, the difference in the time dedicated to gaming according to sex and the frequency of playing a video game according to age. It is a rigorous study, documented, explained and with relevant and recent references within the area.

The length of the paper seems adequate and conforms to what is proposed in the journal's instructions. The structure is also coherent and pertinent, clearly including theoretical aspects and previous studies in the introduction, methodological details of the research, results, discussion and conclusions. 

The title is appropriate and succinctly presents the topic addressed, as well as the abstract. In the case of the latter, a longer sentence is missing in the Methods section with a small description of the persons participating in the study (occasional players), which can be done in a single sentence.

The keyword "academic performance" does not refer to the study. I suggest that it be changed to a more accurate one. The rest of the words are well selected and represent the key issues addressed in the abstract.

The word academic performance been deleted.

Regarding the originality of the study, it is a topic covered in previous studies (which are not overly addressed in the theoretical framework of the text). However, as the authors indicate, it seems that this type of work focuses on other variables (other types of games), and the fact of using an appropriate methodology together with a secondary school sample in a current environment is a strong point of the study. Furthermore, it also seems to be a strong point and an original aspect of the work to analyse the relationship between the two aspects studied (types of games and personality) at this educational stage. This type of study with this sample is difficult to carry out, so it seems clearly a valuable aspect of this research.

As mentioned above, the state of the art on the subject is well constructed, although it includes few studies of relevant and recent quality in the field of research, both nationally and internationally. The design of the study is also rigorous, applying data collection instruments appropriate for the variables proposed to be studied and the possible relationship between the variables considered. The statistical analysis applied is correct and well justified. Regarding the results, they are well described, and the categories help to organize the information on the measures used and the data obtained. However, it is suggested to add a more complete statistical analysis showing some predictive model between game type and personality. That is, how much the personality of the students is explained by the type of game they play.

Also, although some limitations are shown in the study, in my humble opinion other limitations are not adequately included. I suggest supplementing: "However, these findings should be interpreted with caution due to the fact that this study was not interventional but was based on data reported by schoolchildren. Also, undoubtedly, the low sample size is another limitation. Similarly, it is difficult to infer a cause-and-effect relationship between type of play and personality, since, as mentioned there are confounding factors that probably influence these relationships and have not been considered in this study (characteristics of the schoolchild). Therefore, these effects deserve to be further investigated in future studies and shed lighter on this association."

Added

Other than that, this research also not focusing not focusing on analysis between gender and video game playing which could be good research in future.

The low sample size also could be one of the limitations. Similarly, it is difficult to infer a cause-and-effect relationship between type of play and personality, since, as mentioned there are confounding factors that probably influence these relationships and have not been considered in this study (characteristics of the schoolchild). Therefore, these effects deserve to be further investigated in future studies and shed lighter on this association.

For upcoming research, research also can focus on the differences of personality and games their like to play.

In addition, although the conclusions do not highlight the method of the selected study, they do briefly put forward some ideas on prospective that could be completed with some reflection and general documentation on educational strategies in this area. That is, it is suggested to briefly reflect on the extent to which the educational system should take responsibility for the development of students, including a brief review of educational legislation and the role of the institute to prevent and/or treat this type of problems derived from the type of game and the personality of the students. These are some suggestions that can be considered to improve the quality of education in this multicultural context and, especially of interest to teachers, guidance tips and, in general, members of the educational community since this is a journal of students, and considering that the article includes educational/social variables, I believe that a small effort can be made to broaden the educational implications of the study in the sense of what can be done in the future by education to improve the integral development of the students mentioned above.

Added

Educational system should take responsibility for the development of students, including a brief review of educational legislation and the role of the institute to prevent and/or treat this type of problems derived from the type of game and the personality of the students. These are some suggestions that can be considered to improve the quality of education in this multicultural context and, especially of interest to teachers, guidance tips and, in general, members of the educational community since this is a journal of students, and considering that the article includes educational/social variables, I believe that a small effort can be made to broaden the educational implications of the study in the sense of what can be done in the future by education to improve the integral development of the students mentioned above.

Finally, it should be pointed out that the bibliographic references are adequate since they follow the standards required by the journal, both in the text itself and in the final references section. It is also suggested to review some spelling mistakes in the text (example, see Table 1).

PRESENTATION AND ADEQUACY TO THE JOURNAL'S STANDARDS

- Title: It is adequate since it is in accordance with the recommended norms.

- Abstract: It is of a length (221 words) adjusted to what is recommended in the norms.

- Key words: It does not adjust a word to the object of study.

- Length: Complies with the regulations.

- Structure: Conforms to standards.

- Names, symbols and nomenclature: Complies with standards.

- Schemes, drawings, graphs, charts, tables, etc.: Conforms to standards.

- Bibliographical references: Complies with standards. It is suggested to include a study from the children's journal related to the object of this study to complement the quality of the same. 

Round 2

Reviewer 1 Report

I thank the authors for their work and for taking some of my observations into account.

However, I still have three comments to make on this new version of their paper.

Introduction

The introduction has been enriched but it could be reworked on the form to better articulate the different topics covered. In the current state, it is a bit confusing and the sequence of paragraphs and their articulation is not very clear.

An example: the subject of gaming and personality is covered at the beginning lines 33-49, then again at the end, lines 115-123.

I recommend that the authors review their introduction by better articulating the various topics covered: General information on video game practices in the general population, then among adolescents, time spent playing, gaming preferences and finally personality traits.

The paragraph of lines 141_149 would have more place in the conclusion.

Methods

My initial comment on the sample was misunderstood.

“the description of the sample recruitment procedures should be enriched. The limitations in terms of representativeness of this sample should be discussed.”

The point to be addressed was not the sample size but the sample selection procedure.

The authors say:

“Convenient sampling was used as the entire sampling process was carried out in a single step, with each subject independently selected from the other population members. … A total of 420 school students were recruited in this study.”

This is not very clear. Is it possible to specify the method used for recruitment?

On the other hand, the authors state:

“This sampling technique demonstrated good generalization power of the findings to the entire population. “

If it is indeed a “convenient sampling”, it seems to me that this method leads to limits about its ability to generalize. This deserves to be discussed.

Conclusion

I remain convinced that the secondary results of this study (no difference between gender and the hours taken to play games, no difference between age and frequency of gameplay), which are not the subject of an in-depth analysis, impoverish the paper. But I let the authors judge their choice.

However, the conclusion seems a bit short to me. It seems to me that the paragraphs of lines 141-149, 483-484 and 485-500 would have their place in the conclusion.

Author Response

Balan Rathakrishnan, PhD

Associate Professor

Faculty of Psychology and Education

Universiti Malaysia Sabah

Jalan UMS, 88400 Kota Kinabalu

Sabah.

15 February 2023

Editor-in-Chief

Editorial Board

Dear Respected Editor-in-Chief,

We wish to submit an article entitled, Gaming Preferences and Personality among School Students

  1. We would like to confirm that this article has never been published and is not currently under consideration for publication elsewhere.

This research focuses on how gaming preference are related to their personality, gender and the amount of time young adults spend playing a video game. This article explain how personality, gaming preference related to overall health of adolescent especially when someone use excessively use video games among young children and school children.

The content of this manuscript is relevant to the theme of this journal.

I have done all the correction which is given by the reviewers.

Please address all correspondence concerning this manuscript to me at rbhalan@ums.edu.my.

Thank you for your consideration.

Sincerely,

Balan Rathakrishnan, Phd
